# Spitzenkörper assembly mechanisms reveal conserved features of fungal and metazoan polarity scaffolds

Peng Zheng[1], Tu Anh Nguyen [1], Jie Yun Wong[1], Michelle Lee[1], The-Anh Nguyen[1], Jing-Song Fan[2], Daiwen Yang[2] & Gregory Jedd[1,2] ✉

The Spitzenkörper (SPK) constitutes a collection of secretory vesicles and polarity-related proteins intimately associated with polarized growth of fungal hyphae. Many SPK-localized proteins are known, but their assembly and dynamics remain poorly understood. Here, we identify protein-protein interaction cascades leading to assembly of two SPK scaffolds and recruitment of diverse effectors in *Neurospora crassa*. Both scaffolds are transported to the SPK by the myosin V motor (MYO-5), with the coiled-coil protein SPZ-1 acting as cargo adaptor. Neither scaffold appears to be required for accumulation of SPK secretory vesicles. One scaffold consists of Leashin-2 (LAH-2), which is required for SPK localization of the signalling kinase COT-1 and the glycolysis enzyme GPI-1. The other scaffold comprises a complex of Janus-1 (JNS-1) and the polarisome protein SPA-2. Via its Spa homology domain (SHD), SPA-2 recruits a calponin domain-containing F-actin effector (CCP-1). The SHD NMR structure reveals a conserved surface groove required for effector binding. Similarities between SPA-2/JNS-1 and the metazoan GIT/PIX complex identify foundational features of the cell polarity apparatus that predate the fungal-metazoan divergence.

[1] Temasek Life Sciences Laboratory, 1 Research Link, National University of Singapore, Singapore 117604, Singapore. [2] Department of Biological Sciences, National University of Singapore, 16 Science Drive 4, Singapore 117558, Singapore. ✉email: gregory@tll.org.sg

Eukaryotic diversity is manifest in a stunning variety of cellular form and function. From unicellular yeasts to multicellular plants and animals, the ability to polarize signaling, cytoskeleton and endomembrane trafficking underlies the fundamental processes of morphogenesis and differentiation[1–3]. In a given cell type, polarization appears to involve the combined action of ancient functional modules, such as Rho GTPases and cytoskeletal elements, operating under the control of lineage-specific regulatory components[4,5]. Intensively studied in the yeasts, polarization is a sequential process consisting of the selection of a defined cortical site, recruitment of polarity establishment proteins, F-actin polymerization, cytoskeleton-dependent recruitment of secretory and endocytic machineries, and reinforcement of the polarity axis through scaffold assembly and transport-mediated positive feedback (reviewed in refs. [2,3,5]).

In certain cell types, such as neurons and fungal hyphae, persistent vesicle organizing centres assemble at sites of polarization. In the fungi, the Spitzenkörper (SPK) is a phase-dark structure seen by light microscopy at the growing hyphal tip in multicellular Ascomycetes (Pezizomycotina) and Basidiomycetes (Agaricomycotina)[6–8], (reviewed in refs. [9–12]). Electron microscopy reveals an actin filament-containing core[13], which is likely the phase-dark region, surrounded by ~70–100 nm secretory vesicles[14]. In the Pezizomycotina, the core also contains ~40 nm micro-vesicles[14]. Interestingly, macro- and micro-vesicles appear to transport distinct secreted cargoes, suggesting that they comprise different types of post-Golgi secretory vesicles[15,16].

Vesicles are delivered to the SPK through long-range microtubule-mediated transport, followed by short-range transport via type V myosin motors and actin filaments[17–22]. The conserved exocyst complex promotes vesicle fusion with the plasma membrane and also appears to be required for SPK stability[23]. Various polarity-related signaling proteins accumulate at the SPK. These include the nuclear dbf2-related (NDR) kinase COT-1 and associated proteins[24], and the polarisome protein SPA-2[25–27]. In budding yeast, the polarisome comprises Spa2, Pea2, and the F-actin polymerization factors Bni1 and Bud6[28], all of which colocalize at sites of cell growth and are required to maintain proper cell shape. Recent work has identified an additional polarisome component, Aip5, which synergizes with Bni-1 to promotes F-actin polymerization[29]. Spa2 binds to Bni1 and Aip5 through a C-terminal domain[29,30], while its conserved Spa homology domain (SHD) interacts with MAP kinase components[28,31] and Rab GTPase activating (GAP) proteins[32], suggesting that it plays a central scaffolding role. Pea2 is required for SPA-2 tip-localization[33]. However, the precise function of Pea2 is unknown. The majority of characterized SPK proteins are conserved in budding yeast[34], which does not produce a persistent vesicle supply centre. In Neurospora crassa (N. crassa) and Aspergillus nidulans, BUD-6/BudA do not colocalize with SPA-2/SpaA in the SPK[25,27]. Moreover, Pea2 has not been identified outside close relatives of budding yeast[27]. Thus, the role of the polarisome, and determinants of the SPK's unique features remain unclear.

Using the N. crassa model system, we previously identified Spitzenkörper-1 (SPZ-1) as a novel coiled-coil SPK protein present in the SPK-containing multicellular Ascomycota, but absent in budding and fission yeasts[35]. Here, we show that SPZ-1 acts as a cargo adaptor allowing the MYO-5 motor to transport two distinct scaffold complexes to the SPK. One consists of Leashin-2 (LAH-2), which is required for SPK-residency of the signalling kinase COT-1, and the glycolysis enzyme GPI-1. The other is made up of a megadalton hetero-oligomer composed of SPA-2 and Janus-1 (JNS-1). SPA-2 employs its conserved SHD to recruit a novel calponin domain-containing F-Actin effector, CCP-1. The SHD NMR structure reveals a conserved surface groove required

for effector binding. Similar interactions and sequence features of SPA-2/JNS-1 and the mammalian G protein-coupled receptor kinase interacting ArfGAP (GIT)/ p21-activated kinase-interacting exchange factor (PIX) scaffold suggests an ancestral relationship that predates the fungal/metazoan split. By contrast, SPZ-1 and LAH-2 appear to have evolved at key junctures leading to multicellularity in the Ascomycota.

## Results

**Identification of SPZ-1 interacting proteins**. To identify N. crassa SPZ-1 interacting proteins, we employed immunoprecipitation (IP) and mass spectrometry. SPZ-1 co-precipitating proteins include MYO-5, SPA-2, and an uncharacterized protein, NCU03458. Based on its role in SPA-2 transport (see below), we name the latter JANUS-1 (JNS-1) after the Roman god of passages and transition. An epitope-tagged version of each pulls down the others (Fig. 1a and Supplementary Table 1), suggesting that they form a stable complex. SPZ-1 and LAH-2[36] also co-precipitate (Fig. 1a and Supplementary Table 1). All of these proteins possess conserved predicted coiled-coil domains (Fig. 1b), suggesting a basis for their interaction. In keeping with their co-precipitation, mGFP fusions produced from chromosomal loci all localize to the SPK (Fig. 1c). Deletion strains reveal diminished growth rates for Δspz-1, Δspa-2, Δjns-1, and Δlah-2, indicating that each plays an important cellular role (Fig. 1d). To assess their ability to nucleate complex assembly, we ectopically targeted SPZ-1, SPA-2 and JNS-1 to the peroxisome surface (see Methods). Each protein is sufficient to recruit GFP-tagged versions of the others to the peroxisome membrane (Fig. 1e). Moreover, each promotes the aberrant accumulation of peroxisomes at the SPK, indicating that the complexes formed are competent for MYO-5 engagement and transport (Fig. 1f).

**Ordered dependencies leading to SPK-localization**. We next employed sexual crossing to combine the tagged proteins and deletion strains in all possible combinations. These data reveal hierarchical relationships leading to SPK-residency (Fig. 2a). MYO-5 localizes to the SPK independently of all the other proteins. SPZ-1 only depends on MYO-5, while all others depend on SPZ-1. These findings suggest that SPZ-1 acts as a cargo adaptor allowing MYO-5 to transport SPA-2, JNS-1 and LAH-2. Relationships between SPA-2, JNS-1 and LAH-2 are more complex. SPA-2 depends on JNS-1, but JNS-1 retains weak SPK-residency in the absence of SPA-2. LAH-2 also retains weak SPK-localization in the absence of SPA-2 and JNS-1. None of the other proteins depend on LAH-2, suggesting that it is a terminal component of the localization pathway. Together, these data place MYO-5 upstream of SPZ-1, SPZ-1 upstream of SPA-2, JNS-1 and LAH-2, and JNS-1 upstream of SPA-2.

Native polyacrylamide gel electrophoresis (native PAGE) was next used to investigate the formation of complexes by SPZ-1 interacting proteins (Fig. 2b and Supplementary Fig. 1). MYO-5 and SPZ-1 both migrate at approximately 800 kDa and their banding patterns are unaffected in the absence of the other proteins (Supplementary Fig. 1). By contrast, the banding patterns of JNS-1 and SPA-2 are interdependent (Fig. 2b). Both migrate as three bands of approximately 600, 900 and 1100 kDa. When either JNS-1 or SPA-2 is absent, the larger species of the other collapse to the 600 kDa band. This molecular weight is significantly higher than the ~100 kDa predicted molecular weights of SPA-2 and JNS-1, suggesting that they may both form homo-oligomers. To examine this possibility, we used heterokaryons to combine mGFP- and HA-tagged versions in the deletion background of the other. For both proteins, when precipitation is carried out with anti-GFP antibodies, HA-tagged versions are co-precipitated (Fig. 2c).

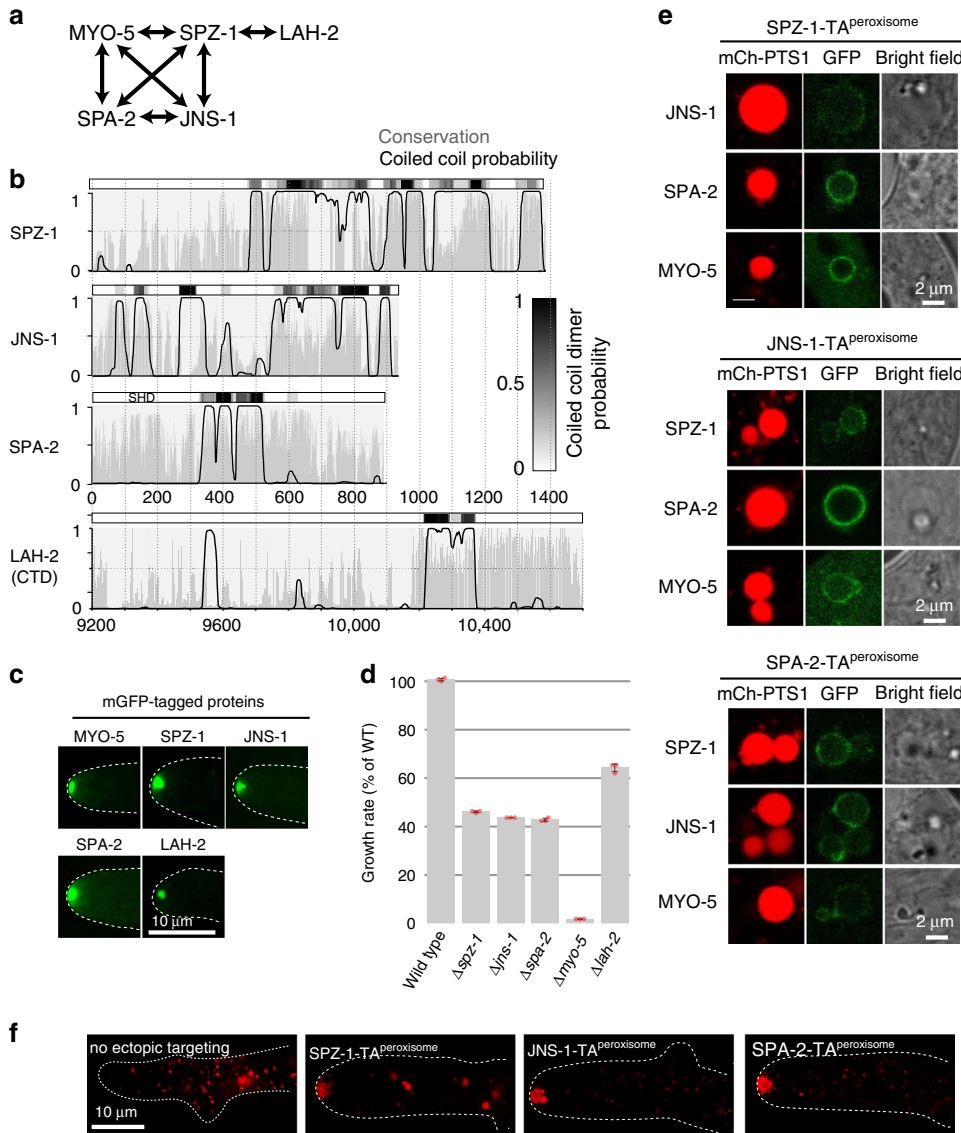

**Fig. 1 Identification of the SPZ-1 interacting proteins. a** The indicated proteins were HA-epitope tagged at their endogenous loci and co-precipitating proteins were identified by mass spectrometry. Double arrows identify mutual co-precipitators. Details are provided in Supplementary Table 1. **b** Sequence conservation (grey) and coiled-coil probability (black) of the indicated proteins. The horizontal bar above each graph shows the coiled-coil dimer probability according to the greyscale shown in the legend. **c** The localization of chromosomally-encoded mGFP fusion proteins is shown for the indicated proteins. Dotted white lines show the hyphal outline. Scale bar = 10 μm. **d** For the indicated strains, the average growth rate is shown as mean values ± SD ($n = 3$ independent measurements shown as opaque red dots). **e** The indicated HA-epitope tagged proteins were fused to a C-terminal peroxisome tail-anchor (TA$^{peroxisome}$) and combined through sexual crosses with the indicated mGFP-fusion proteins. Under this condition, SPZ-1, JNS-1 and SPA-2 can each recruit the others to the surface of the peroxisome. mCherry-PTS1 provides a marker of the peroxisome matrix. Scale bar = 2 μm. **f** Aberrant peroxisome accumulation at the SPK shows that ectopically assembled complexes are all competent for transport. The no ectopic panel shows the normal distribution of peroxisomes. Dotted white lines show the hyphal outline. Scale bar = 10 μm. Source data are provided as a Source Data file.

Together, these data show that SPA-2 and JNS-1 form homo-oligomers, which further associate to form two stable hetero-oligomeric species (Fig. 2b). The intimate relationship between JNS-1 and SPA-2 is further demonstrated by their dependency on one-another for pull-down by SPZ-1 (Fig. 2d). For JNS-1, this does not appear to be consistent with its weak SPK-localization in the SPA-2 mutant, which is presumably dependent on SPZ-1. Steady-state levels of JNS-1 appear to be diminished in the SPA-2 mutant (Fig. 2b), suggesting that it requires SPA-2 for stability. Thus, in the absence of SPA-2, JNS-1's SPK-localization may be the result of its aberrant interactions. Alternatively, weak binding to SPZ-1 may not be captured by IP.

**Two functionally distinct domains in SPZ-1, SPA-2 and JNS-1.** Data presented thus far show that SPZ-1 acts as a cargo adaptor allowing MYO-5 to transport SPA-2/JNS-1 and LAH-2 to the SPK. To investigate the basis for these interactions, we deleted discrete regions of SPZ-1 selected based on a combination of coiled-coil prediction and evolutionary conservation (Fig. 3a). Deletions were constructed at endogenous loci by replacing selected coding sequences with an in-frame mCherry-selectable marker fusion cassette[37]. The resulting variants were analysed for loss-of-function, localization, interaction, and steady-state protein levels. This analysis identifies two key regions whose absence leads to distinct SPZ-1 loss-of-function phenotypes (Fig. 3b, c and

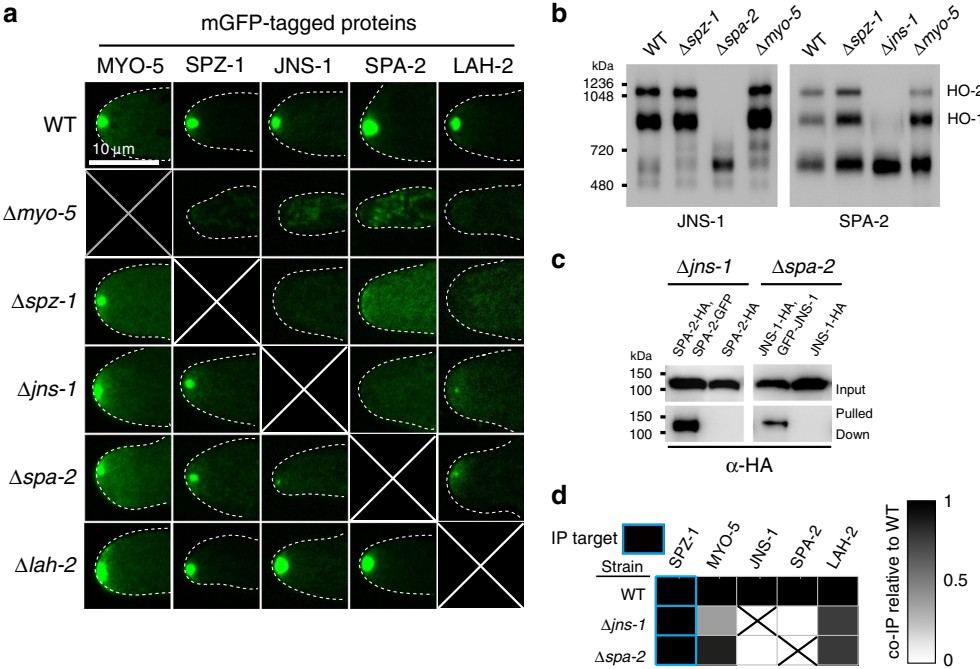

**Fig. 2 SPK-localization dependencies and complex formation. a** The localization of the indicated mGFP-tagged proteins was determined by confocal microscopy in the indicated strains. Dotted white lines show the hyphal outline. Scale bar = 10 µm. **b** Native PAGE identifies two stable hetero-oligomeric complexes formed by JNS-1 and SPA-2 (HO-1 and HO-2). This figure is related to Supplementary Fig. 1. **c** SPA-2-mGFP precipitates SPA-2-HA in the absence of JNS-1 (left panels), and JNS-1-mGFP precipitates JNS-1-HA in the absence of SPA-2 (right panels). **d** SPA-2 and JNS-1 depend on each other for interaction with SPZ-1. The signal from mass spectrometry is compared to the wild type (WT) precipitation according the scale shown in the legend. Source data are provided as a Source Data file.

Supplementary Fig. 2a). Deletion of the highly conserved region 3 coiled-coil domain abolishes the ability of SPZ-1 to precipitate SPA-2/JNS-1 and LAH-2. However, MYO-5 interaction (Fig. 3c) and SPK-localization (Fig. 3b) are retained, albeit at somewhat diminished levels, possibly due to diminished steady-state accumulation of this variant as compared to wild-type SPZ-1 (Supplementary Fig. 2a). By contrast, the coiled-coil region 6 deletion variant retains cargo binding, but abolishes MYO-5 binding (Fig. 3c) and SPK-localization (Fig. 3b), indicating that it is responsible for motor engagement. These data show that SPZ-1 binds to cargos and MYO-5 through distinct coiled-coil domains.

JNS-1 and SPA-2 can also be dissected into two discrete functional regions (Fig. 3d, g and Supplementary Fig. 2b,c). N- and C-terminal coiled-coil domains of JNS-1 are essential for SPK-residency (Fig. 3e). However, only the N-terminal domain is required for hetero-oligomer formation with SPA-2 (Fig. 3f). SPA-2 possesses an essential coiled-coil domain and neighbouring sequences (regions 3 and 4), required for hetero-oligomer formation with JNS-1 (Fig. 3h, j) and SPK-residency (Fig. 3i). By contrast, deletion of the N-terminal SHD containing region 1 results in loss-of-function, but does not impair SPK-residency or hetero-oligomer formation with JNS-1, suggesting its exclusive association with effector recruitment. IP of variants corroborates these conclusions: the SPA-2 region 3/4 deletions impair precipitation with JNS-1, SPZ-1 and MYO-5, while the region 1 SHD deletion retains this ability (Fig. 3j).

**NMR structure of the SPA homology domain.** The SHD was originally identified as a direct repeat conserved between the mammalian polarity scaffold protein GIT and yeast Spa2[38]. Because of its conserved and apparently central role in effector recruitment, we purified the *N. crassa* SHD and determined its NMR solution structure (Fig. 4 and Supplementary Fig. 3, PDB ID: 6LAG). The overall fold consists of six alpha-helical segments

(Fig. 4b). The conserved direct repeats encode α-2 and α-3 (repeat 1), and α-4 and α-5 (repeat 2) (Fig. 4b, c). Conserved residues in these segments form a surface groove with a partially hydrophobic base and positively charged rims. Antiparallel arrangement of α-3 and α-5 form the groove base, while antiparallel α-2 and α-4 form the rims (Fig. 4b–d). In the mammalian SHD, the L288A mutation abolishes binding to Piccolo and FAK, but not to GIT[39]. Sequence alignment shows that L288 is conserved in the *N. crassa* SHD (L133) where it contributes hydrophobicity to the groove's base (Fig. 4c, d). The L133A mutation in *N. crassa* SPA-2 leads to a full loss-of-function (Supplementary Fig. 4a), suggesting that fungal and metazoan SHD domains recruit effectors through a similar structural moiety.

**Identification of a new SHD effector.** Initial SPA-2 IP experiments did not identify *N. crassa* SHD effectors. We reasoned that this might be due to interference of detergents with binding to the SHD groove. Indeed, when detergent is excluded from IP washes, an uncharacterized protein (NCU00277) co-precipitates with SPA-2 (Fig. 5a, b). NCU00277 contains an N-terminal calponin homology domain and central coiled-coil domain (Fig. 5c). We therefore named it calponin coiled-coil protein-1 (CCP-1). The L133A mutation significantly diminishes the ability of SPA-2 to bind CCP-1, indicating that they interact through the SHD (Fig. 5a, b). The calponin homology domain and coiled-coil region are both essential for CCP-1 function and localization to the SPK (Fig. 5c). Calponin homology domains occur in diverse actin regulatory proteins[40]. Thus, we next examined the impact of *ccp-1* deletion on F-actin distribution. Loss of CCP-1 and its upstream regulators, lead to significantly diminished levels of SPK F-actin (Fig. 5d, e). Interestingly, CCP-1 loss-of-function also impairs SPK-incorporation of LAH-2, but does not affect SPK-residency of SPZ-1, SPA-1 or JNS-1 (Fig. 5f). Together, these data

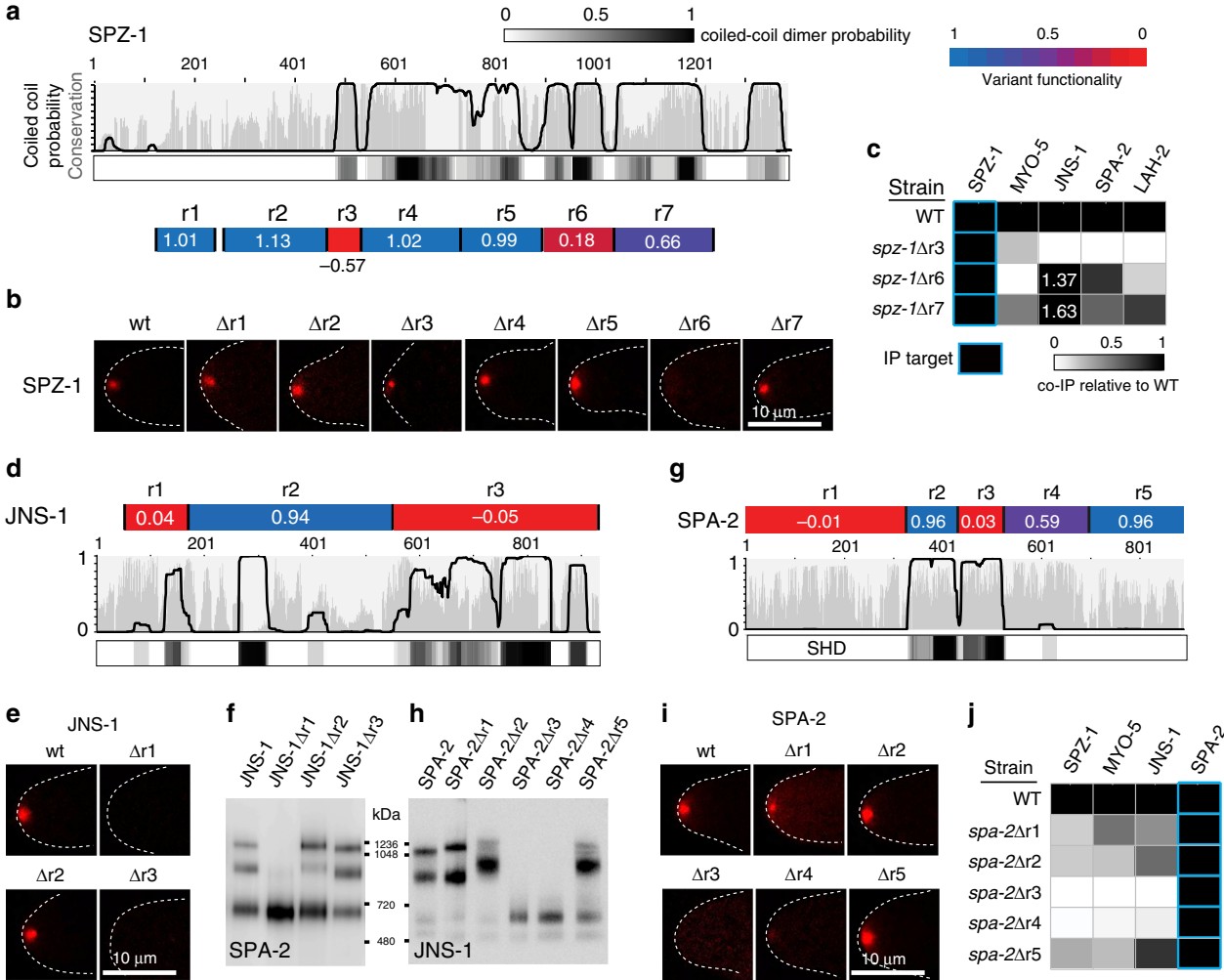

**Fig. 3 Functional dissection of SPZ-1, JNS-1 and SPA-2. a** Conservation, coiled-coil, and coiled-coil dimer probability is shown for SPZ-1. The indicated regions of SPZ-1 (r1–r7) were replaced by an mCherry-selectable marker fusion as described in materials and methods. Functionality of the variants is scored according to the color scale, where 1 and 0 represent wild type and deletion mutant growth rates, respectively. This figure is related to Supplementary Fig. 2a. **b** Localization of the indicated SPZ-1 deletion variants. Dotted white lines show the hyphal outline. Scale bar = 10 μm. **c** The ability of the indicated SPZ-1 variants to co-precipitate interacting proteins is compared to full-length SPZ-1 (WT). The signal from mass spectrometry is compared to WT according the scale shown in the legend. If the signal from mass spectrometry exceeds that of the WT, this value is identified with white numbers. **d** Conservation, coiled-coil and coiled-coil dimer prediction for JNS-1. The indicated regions of JNS-1 (r1-r3) were replaced by an mCherry-selectable marker fusion as described in materials and methods. Variant functionality is indicated as in **a**. This figure is related to Supplementary Fig. 2b. **e** Localization of the indicated JNS-1 deletion variants. Dotted white lines show the hyphal outline. Scale bar = 10 μm. **f** Native PAGE analysis of SPA-2 mobility in the indicated JNS-1 deletion variants. **g** Conservation, coiled-coil and coiled-coil dimer probability for SPA-2. The indicated regions of SPA-2 (r1–r5) were replaced by an mCherry-selectable marker fusion as described in materials and methods. Variant functionality is indicated as in (**a**). This figure is related to Supplementary fig. 2c. **h** Native PAGE analysis of JNS-1 mobility in the indicated SPA-2 deletion variants. **i** Localization of the indicated SPA-2 deletion variants. Dotted white lines show the hyphal outline. Scale bar = 10 μm. **j** The ability of the indicated SPA-2 variants to co-precipitate interacting proteins is compared to full-length SPA-2 (WT). The mass spectrometry is quantified as in (**c**). Source data are provided as a Source Data file.

suggest that CCP-1 participates in transport-mediated positive feedback to stabilize SPK F-actin.

**Identification of LAH-2 effectors**. Data presented thus far identify an ordered cascade of protein-protein interactions leading to assembly and SPK-residency of two distinct SPK scaffolds. The SHD domain of the JNS-1/SPA-2 complex recruits the actin effector CCP-1 to the SPK (Fig. 5). By contrast, the role of LAH-2 remains unclear. We therefore screened proteins known to reside at the SPK for LAH-2 dependency (Fig. 6a). This identified the polarity-associated NDR kinase, COT-1 and its regulatory binding partner MOB-2A[24] as LAH-2 clients. In unrelated work we found that the glycolysis enzyme glucose-6-phosphate isomerase

(GPI-1) is localized to the *N. crassa* SPK. It also depends on LAH-2. All three proteins depend on SPZ-1, and like LAH-2 show diminished SPK-residency in the absence of SPA-2 and JNS-1 (Fig. 6a).

**Scaffold mutants are not impaired in vesicle accumulation**. Scaffold clients identified here are associated with signaling (COT-1/MOB-2A), actin regulation (CCP-1), and metabolism (GPI-1). However, none appears to be directly associated with vesicle trafficking. We concluded this study by examining the effect of scaffold loss-of-function on secretory markers. Remarkably, SPK-localization of markers associated with post-Golgi micro- (CHS-1) and macro-vesicles (GS-1)[15] as well as markers of early- (YPT-1)

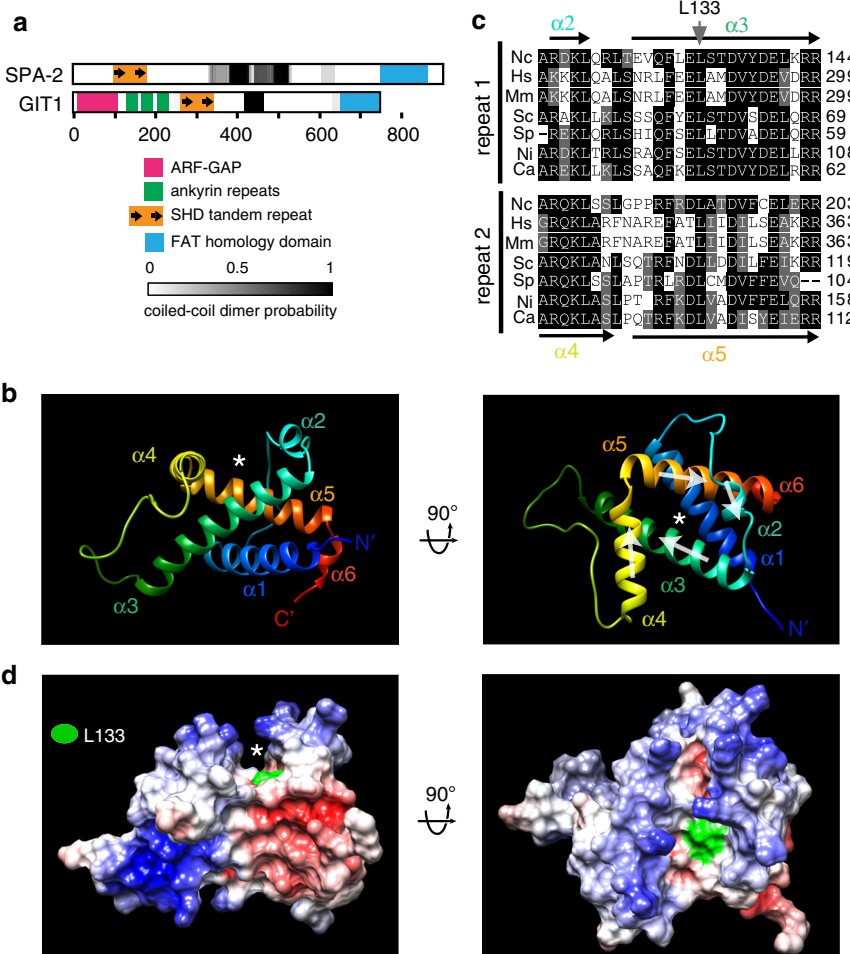

**Fig. 4 Solution structure of the *Neurospora crassa* SPA-2 SHD (Spa homology domain). a** The cartoon shows the position of domains in *N. crassa* SPA-2 and Human GIT1. Domains are identified according to the legend. **b** The SHD structure (G84-E211) is shown in a rainbow-colored ribbon diagram. Alpha-helical segments and N- and C-terminal ends are labeled. The right panel shows the structure after the indicated rotation. The N- to C-terminal directionality of selected helices is indicated with opaque white arrows. The surface groove whose rims are formed by antiparallel α-2 and α-4 is identified with an asterisk. **c** The SHD tandem repeat sequences from representative metazoan and fungal SHDs are aligned. Identical residues are shaded black and conserved residues grey. Alpha helices are labeled according to colors shown in **c**. The L133 residue associated with effector binding is identified with a grey arrow. *Neurospora crassa* (Nc), *Homo sapiens* (Hs), *Mus musculus* (Mm), *Saccharomyces cerevisiae* (Sc), *Schizosaccharomyces pombe* (Sp), *Neolecta irregularis* (Ni) and *Candida albicans* (Ca). **d** The SHD electrostatic surface projection reveals positive charge at the groove's rims. The perspective of the two panels are the same as shown in (**b**). The surface groove is identified with an asterisk. The L133 residue is shown in green. This figure is related to Supplementary Fig. 3.

and late-Golgi/post-Golgi vesicles (YPT-31 and SYN-1), do not appear to be significantly altered in the mutants (Fig. 6b). Moreover, the inner and out layers of the SPK also appear to form normally (Fig. 6c). Thus, while MYO-5 is known to be required for delivery of post-Golgi secretory vesicle delivery to the *N. crassa* SPK[21], the scaffolds identified here do not appear to be directly related to this process.

## Discussion

Cell polarity requires the coordinated regulation of signaling, cytoskeletal dynamics, and membrane trafficking. Protein scaffolds act as points of convergence to organize these diverse activities. However, an overall understanding of these complex systems is lacking. Here, we characterize the assembly of Spitzenkörper scaffold complexes and effectors associated with F-actin reinforcement (CCP-1), signalling (COT-1) and metabolism (GPI-1). SPZ-1 plays a key role as cargo adaptor allowing MYO-5 to promote SPK-residency of the ancient polarisome-related SPA-2/JNS-1 complex and the Pezizomycotina-specific LAH-2 scaffolds. Neither appears to be associated with MYO-5 dependent transport

of vesicles to the SPK (Fig. 7). This has important implications, suggesting that parallel transport pathways can allow for SPK-specific regulatory interactions between scaffold clients and vesicles (see below).

Scaffold effectors support distinct activities associated with cell polarity. CCP-1 loss-of-function leads to significantly diminished SPK F-actin, suggesting that it acts as part of an transport-mediated positive feedback loop (Fig. 7a). Such a role for CCP-1 is consistent with findings in metazoan systems where non-muscle calponin proteins stabilize F-actin networks[41,42]. COT-1 is a member of the ancient NDR kinase family and plays an essential role in cell polarity[43]. By contrast, LAH-2 is non-essential, suggesting that its scaffold function plays a regulatory role to promote COT-1 activity. GPI-1 catalyses the second step in glycolysis. Its association with the SPK suggests coordination between metabolism and tip-growth. Resolving the functional consequence of this intriguing association will require more work.

Several findings support a model in which overall SPK assembly occurs through the concerted action of independent functional modules. The SPK-residency of LAH-2 depends on

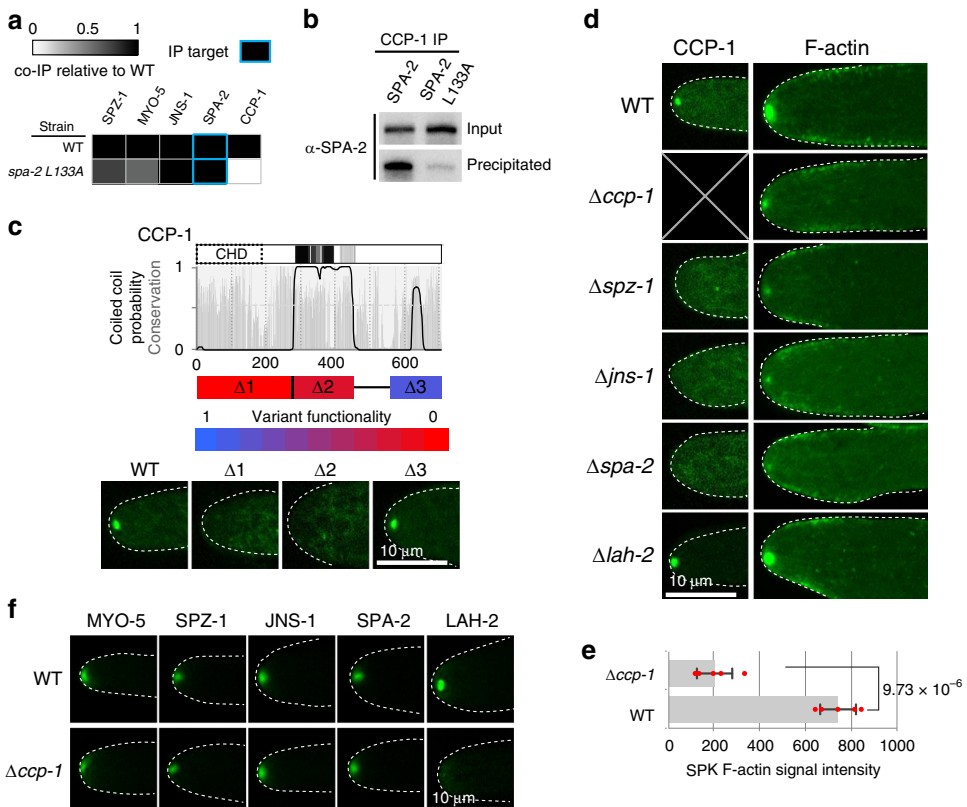

**Fig. 5 Identification of the *Neurospora crassa* SHD binding effector CCP-1. a** The ability of the SPA-2L133A mutant to co-precipitate interacting proteins is compared to full-length SPA-2 (WT). The signal from mass spectrometry is compared to the WT precipitation according the scale shown in the legend. **b** CCP-1-mGFP precipitates SPA-2, but interacts weakly with the SPA-2 L133A mutant. Input and precipitated fractions are analyzed by western blotting for SPA-2. **c** Conservation, coiled-coil and coiled-coil dimer prediction for CCP-1. The calponin domain is boxed with a dashed line. The indicated regions of CCP-1 (r1-r3) were deleted using an mCherry-selectable marker fusion as described in Materials and Methods. Variant functionality is scored according to the color scale where 1 and 0 represent wild type and deletion mutant growth rates, respectively. Lower panels show localization of the indicated deletion variants. Dotted white lines show the hyphal outline. Scale bar = 10 μm. **d** CCP-1 tip-localization depends on SPZ-1, JNS-1 and SPA-2, but not LAH-2. The distribution of F-actin is shown for the indicated strains. Dotted white lines show the hyphal outline. Scale bar = 10 μm. **e** SPK F-actin signal intensity (arbitrary units) is quantified for wild type and *ccp-1* deletion strains. Data are shown as mean values ± SD (n = 5 independent measurements shown as opaque red dots). The p-value calculated from 2-sided Student's t-test is indicated. **f** LAH-2 depends on CCP-1, but other scaffold components do not. Localization of the indicated GFP-tagged proteins is shown in wild type and *ccp-1* deletion mutant. Dotted white lines show the hyphal outline. Scale bar = 10 μm. Source data are provided as a Source Data file.

SPA-2/JNS-1 (Fig. 2a). However, this relationship is not reciprocal. Loss of the SPA-2 effector CCP-1 also leads to diminished SPK-localization of LAH-2, but does not affect SPA-2/JNS-1. Together, these results support a model in which LAH-2's dependence on SPA-2/JNS-1 is an indirect consequence of CCP-1's absence at the SPK and a resulting diminishment in SPK F-actin. This may also be true of BNI-1 which displays similar dependencies as LAH-2 (Supplementary Fig. 4b). The apparently normal stratification of SPK secretory markers in scaffold deletion mutants (Fig. 6b, c) further attests to modular SPK assembly and the independent accumulation of scaffolds and vesicles, as well as differential sensitivity of SPK constituents to levels of SPK F-actin.

Scaffold deletion mutants retain residual levels of SPK F-actin (Fig. 5d) and BNI-1 (Supplementary Fig. 4b). The ability of Formin proteins like BNI-1 to nucleate F-actin polymerization is well-established[44], and in *N. crassa* BNI-1 is known to be an effector of RHO-1 and its nucleotide exchange factor LRG, which localize to the SPK and cell cortex, respectively[45]. Thus, normal levels of SPK F-actin appear to be the product of the converging activities of BNI-1 controlled polymerization, originating from the cell cortex, and CCP-1 mediated stabilization, which depends on MYO-5 transport from sub-apical regions of the hypha.

A number of observations support an ancestral relationship between SPA-2/JNS-1 and the mammalian GIT/PIX polarity scaffolds. The SHD was first identified as a sequence repeat shared by metazoan GIT and yeast Spa2 proteins[38,39,46]. Previous work showed that Spa2 and GIT recruit a variety of effectors through this domain[31,32,39]. Here we show that as with GIT and PIX[39,47], *N. crassa* SPA-2 and JNS-1 form homo-oligomers (Fig. 2c) which further associate to produce a mega-dalton hetero-oligomer (Figs. 2b and 3). The SHD structure shows how the direct repeat contributes to the formation of an amphipathic surface groove (Fig. 4). A mutation previously shown to abolish GIT1 binding to its effectors Piccolo and FAK[39], contributes hydrophobicity to the groove's base, and the corresponding mutation in *N. crassa* SPA-2 disrupts interaction with its effector, CCP-1 (Fig. 5a, b). Thus, fungal and metazoan SHDs employ a similar fold to associate with effectors. An ancestral relationship between SPA-2 and GIT is further supported by significant sequence similarity between the C-terminal GIT focal adhesion targeting domain (FAT, also known as paxillin-binding domain) and C-terminal domains of fungal SPA-2 proteins (Supplementary Fig. 6). However, we note that deletion of the SPA-2 FAT domain does not produce significant loss-of-function (Fig. 3g), suggesting that it plays a minor role in *N. crassa*.

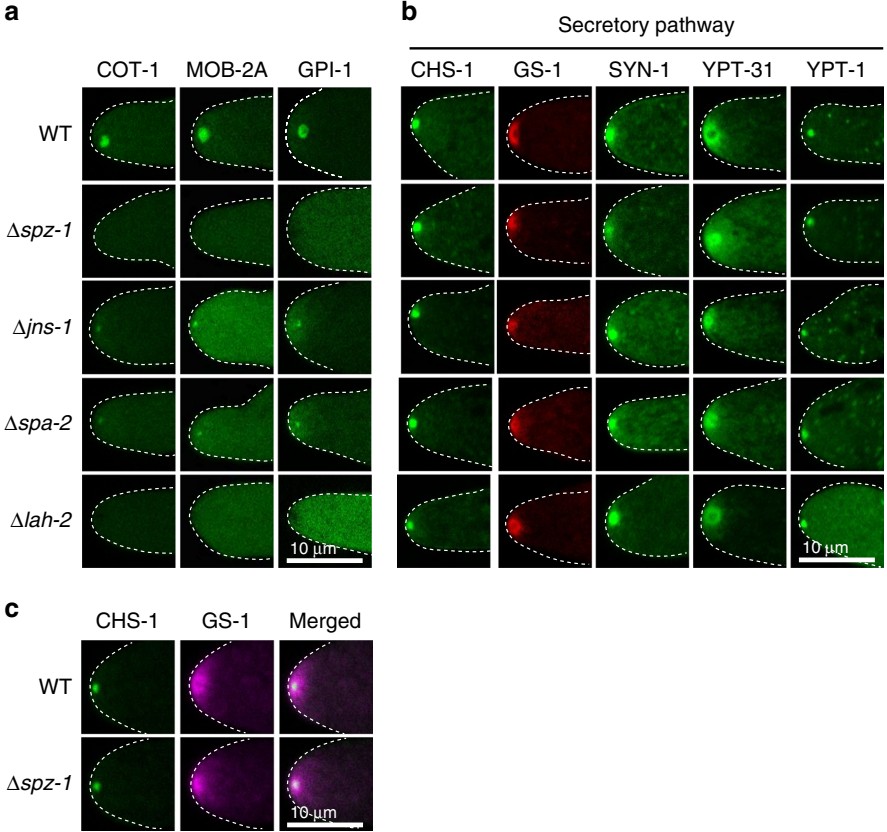

**Fig. 6 Identification of LAH-2 effectors, and the influence of scaffolds on secretory markers. a** COT-1, MOB-2A and GPI-1 depend on LAH-2 for tip-localization. The indicated mGFP fusion proteins were combined with the indicted deletion strains and imaged by confocal microscopy. Note that the partial SPA-2 and JNS-1 dependence of the LAH-2 effectors is similar to that displayed by LAH-2 itself (see Fig. 2a). Dotted white lines show the hyphal outline. Scale bar = 10 μm. **b** Secretory vesicles and organelles do not depend on any of the scaffolds for SPK-localization. The indicated mGFP or mCherry fusion proteins were combined with the indicted deletion strains and imaged by confocal microscopy. Dotted white lines show the hyphal outline. Scale bar = 10 μm. **c** Co-expression of a marker of micro-vesicles (CHS-1) and macro-vesicles (GS-1) in the *spz-1* deletion mutant reveals localization to inner and outer layers of the SPK, respectively. Wild-type (WT) accumulation is shown for comparison. Dotted white lines show the hyphal outline. Scale bar = 10 μm. Source data are provided as a Source Data file.

JNS-1 possesses two essential coiled-coil domains. The N-terminal region 1 is required for complex formation with SPA-2, while both region 1 and the C-terminal region 3 are required for SPK-localization (Fig. 3d–f). These data suggest that JNS-1 has a primary function in promoting SPA-2 accumulation at the SPK. Despite an absence of primary sequence similarity, several commonalities suggest that budding yeast Pea2 performs an analogous function to JNS-1. Both possess predicted coiled-coil domains, and as with *N. crassa* SPA-2 and JNS-1, yeast Spa2 depends on Pea2 for localization to the bud tip[33], Spa2 and Pea2 co-sediment in a large complex (12S by velocity sedimentation)[28], and as with JNS-1, steady-state levels of Pea2 are diminished in the absence of Spa2[33]. Furthermore, in yeast, Spa2 co-precipitates with Myo2 and depends on it for tip-localization[48]. Whether the Myo2 interaction is direct or requires Pea2 or another intermediary remains unclear. Nevertheless, combined with findings presented here, these observations suggest that SPA-2 proteins generally require coiled-coil binding partners and Myosin V motors for polarized accumulation. Future work can address whether a similar relationship exists with the GIT/PIX complex.

Surface features and dimensions of the SHD groove suggest a potential to bind an amphipathic alpha-helical segment through hydrophobic contacts with the base and charged interactions with the rim (Fig. 4). GIT forms a parallel dimer through its coiled-coil domain[47]. This arrangement positions two SHDs to potentially bind clients cooperatively. Interestingly, the GIT SHD linked to the dimerization domain interacts with piccolo more strongly than monomeric SHD[39], suggesting that this may indeed be the case. Alternatively, the dimeric arrangement of SHDs could be exploited to promote interaction between distinct SHD-bound clients. In the future, powerful haploid genetics of *N. crassa* can be used to investigate how the oligomeric presentation of SHD influences its activity.

In metazoans, GIT and PIX regulate diverse polarity-related processes that include focal adhesion dynamics and cell migration[49–51], organ development[52–54], and synapse formation[55] and dynamics[56–58]. From the perspective of domain organization, GIT/PIX are significantly more complex than their fungal counterparts. In addition to the coiled-coil, SHD and FAT domain, GIT proteins contain Arf GAP and Ankyrin repeat domains, while PIX contains calponin homology, SH3, Rho GEF, and PH domains[59]. These domain gains are likely to reflect complexification of polarity signaling during metazoan evolution. Despite this, a dual function in promotion of F-actin polymerization and regulation of membrane trafficking appears to be a conserved feature of fungal and metazoan polarisome-related scaffolds. In metazoans, regulation of F-actin dynamics is achieved through a complex interplay between activation of Rac1 and Cdc42 through the PIX Rho GEF domain and recruitment of the RAC/CDC42 effector PAK via the PIX SH3 domain[59,60]. By contrast, in *N. crassa*, F-actin regulation is achieved through the SPA-2 client CCP-1 (Fig. 5), while in yeast Bni1, Bud6 and Aip5 are involved[28,29].

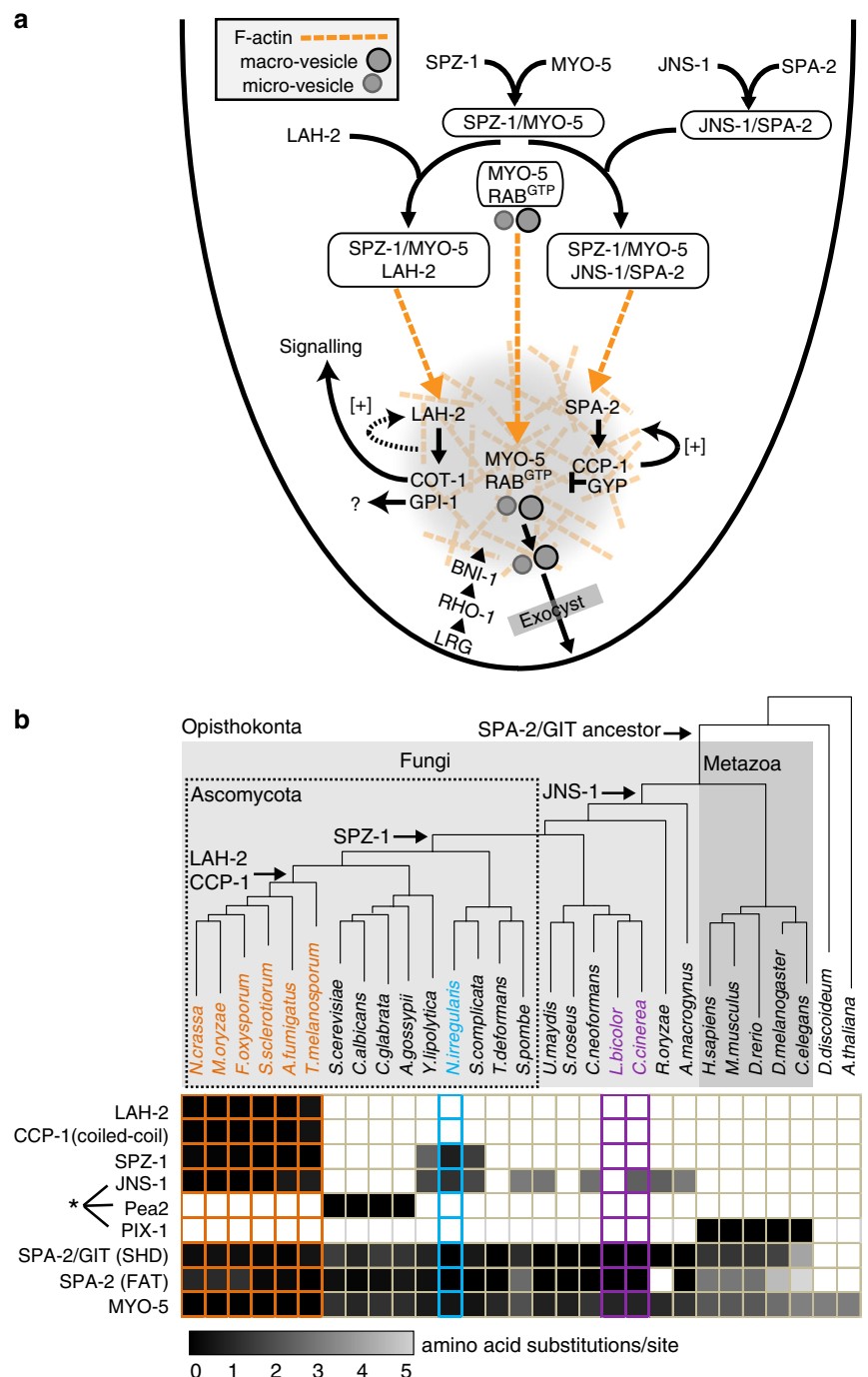

**Fig. 7 A blueprint for assembly of the of SPK proteinaceous scaffold. a** The cartoon depicts scaffold assembly, MYO-5 dependent transport and effector recruitment. Parallel transport of secretory vesicles, and scaffold complexes is indicated with dashed orange arrows. Complexes are boxed. Secretory vesicles and F-actin are depicted according to the legend. The positive feedback from CCP-1 to F-actin is shown with a solid arrow from CCP-1 to an F-actin filament. The dependency of LAH-2 on CCP-1 is likely to be an indirect consequence of diminished SPK F-actin. This relationship is depicted with a dashed arrow from an F-actin filament to LAH-2. See discussion for additional information. **b** Phylogenetic distribution of proteins examined in this study. Filled and empty squares indicate the presence and absence of the indicated protein, respectively. Arrows to branch nodes identify the likely origin of the indicated proteins. Note that with a few exceptions, all the SHD-containing proteins co-occur with coiled-coil binding partners (asterisk). Squares with colored borders identify complex multicellular fungi: the Pezizomycotina (orange), Neolecta (blue) and the Agaricomycotina (purple). Amino acid substitution per site is shown according to the indicated greyscale, which represents the degree of divergence from the reference Pezizomycotina sequences (see Methods). In the case of Pea2 and PIX the reference group is the Saccharomycotina and Metazoans, respectively.

Interestingly, mammalian PIX contains a calponin domain, while in *N. crassa* and presumably other filamentous fungi, the calponin domain in CCP-1 is recruited through the SHD (Fig. 5).

With respect to regulation of membrane trafficking, in the GIT/PIX complex, the GIT ARF-GAP domain[61] influences a variety of membrane trafficking events at the plasma membrane[62,63]. In yeast, the Spa2 SHD binds two related Rab GTPase activating proteins (GAPs), Msb3 and Msb4[32], which display GAP activity towards Sec4[64]. In budding yeast, Myo2 transports post-Golgi secretory vesicles through sequential association with activated

Rab GTPases, Ypt31/32 and Sec4[65,66]. Neurospora encodes a single Msb homolog, GYP-3, which was not captured in our pull-down experiments. However, recent work has shown that it indeed depends on SPA-2 for SPK-residency[67] (Supplementary Fig. 4d). Furthermore, GYP-3 SPK-residency is also abolished in the SPA-2 L133A mutant (Supplementary Fig. 4e), indicating that recruitment occurs through the SHD. N. crassa MYO-5 has been shown to deliver post-Golgi secretory vesicles to the SPK[21]. However, scaffolds identified here do not appear to play a role in vesicle transport (Fig. 6b). These observations lead to a model in which MYO-5/SEC-4 tethered post-Golgi vesicles are likely to encounter high concentrations of SPA-2 clients such as GYP-3 only after delivery to the SPK. Such an arrangement could ensure that the link between vesicle and motor is only terminated in close proximity to the site of exocytosis.

In budding yeast, polarisome loss-of-function leads to abnormally shaped and enlarged cells, suggesting a primary function in morphogenesis[2]. By contrast, in N. crassa polarisome loss-of-function leads to diminished growth rate (Fig. 1d). An analysis of hyphal shape reveals minor defects in morphology as compared to loss-of-function in the endocytic component coronin (Supplementary Fig. 7), which affects both growth rate and morphogenesis[68]. Filamentous fungi such as N. crassa can display remarkably high rates of tip-growth, which can approach 1 μm per second[69] and unlike yeast, morphogenesis and nuclear division are not coupled. Thus, the polarisome is likely to have diverged at the level of effectors to accommodate differing exigencies in hyphal fungi and budding yeast. This idea is further supported by work in Ashbya gossypii and N. crassa, where SPA-2 localization transitions from a yeast-like tip crescent to a SPK sub-apical dot as the rate of hyphal tip growth increases[27,70].

To better understand the evolutionary history of SPK components, we searched for related sequences in representative fungal and metazoan proteomes (Fig. 7b). SPA-2 SHD and FAT domains identify relatives throughout the fungi and metazoans. JNS-1, Pea2 and PIX distribution suggests that SPA-2/GIT proteins generally require a coiled-coil binding partner. Together, these findings point to an ancestral hetero-oligomeric polarisome scaffold that predates the divergence of fungi and metazoans. The analysis further implicates the serial advent of new protein functions in SPK evolution. SPZ-1 maps to the origin of the Ascomycota and is conserved in multicellular filamentous species, but was apparently lost in budding and fission yeast as they independently underwent simplification[35]. Its role in SPK complexification is supported by a dual function in transporting ancient (SPA-2/JNS-1), and Pezizomycotina-specific (LAH-2), scaffold components. A number of innovations appear to have been fixed prior to radiation of the Pezizomycotina. CCP-1 plays a fundamental role in maintaining F-actin in the SPK core and it acquired a functionally important coiled-coil domain at this juncture (Fig. 7b). LAH-2 is related to the tether linking Woronin body septal pore-plugging organelles to the cell cortex[71]. Its role as scaffold suggests co-option to assume additional functionality in the SPK. The advent of several genes required to build Woronin bodies also maps to the Pezizomycotina common ancestor[35]. Thus, the emergence of new proteins leading to integrated novelties in organelle function and cell polarity appears to have preceded a transition to complex multicellular organization and extensive evolutionary radiation.

## Methods

**N. crassa growth, genetic manipulation and microscopy.** N. crassa strains were grown in synthetic Vogel's N (VN)[72]. N. crassa deletion mutants were obtained from the Fungal Genetics Stock Centre[73]. Crosses were performed as previously described[74]. Growth rates were measured using the race tube method[69]. Here, conidia were germinated on solid VN medium and grown at 30 °C overnight. An

agar block containing live hyphae was then excised from the growth front and inoculated on a race tube containing VN medium. The growth rate was measured after incubation of race tubes for two days at 30 °C. Epitope and fluorescent protein tags and partial deletions were generated using Marker Fusion Tagging[37]. The strains employed in this study can be found in Supplementary Data 1.

For the functional dissection of proteins by MFT, deletion variants were selected based on a combination of protein sequence conservation[35], coiled-coil probability and dimer probability as predicted by MultiCoil[75]. Precise deletion breakpoints are identified in the strain genotypes found in Supplementary Data 1. Ectopic targeting to the peroxisomal surface (Fig. 1e,f) was accomplished by appending the tail-anchor from N. crassa PEX-26 to the C-terminus of the indicated proteins. All tagged and mutant strains were backcrossed with wild-type strains FGSC465 or FGSC466 to obtain homokaryons. Additional crosses were then conducted to combine various tagged and mutant strains.

Images of hyphal tips were obtained using a Leica SP8 inverted confocal microscope with the HCX PL APO 100 /1.40 OIL objective. Primary peripheral tips were identified and watched to ensure that they were growing. Images of five tips were taken for each strain and representative tips are shown. Images were exported using ImageJ (http://rsb.info.nih.gov/ij/) and converted into figures with Adobe Illustrator (Adobe Illustrator CS6). Quantitation of SPK F-actin in wild type and the ccp-1 deletions strain (Fig. 5e), was conducted using the linescan function of ImageJ.

**Immunoprecipitation (IP) and mass spectrometry.** To prepare N. crassa extracts, mycelium was grown in liquid VN and harvested, washed and ground to a fine powder in liquid nitrogen as previously described[71]. Extracts were prepared by adding 10 ml of IP buffer (20 mM HEPES pH 7.4, 150 mM KOAc, 2 mM MgCl₂, 1 mM DTT, 0.5 mM PMSF and protease inhibitor cocktail (Roche, 04693159001) to an equal volume of frozen powder. Extraction was conducted with end-over-end inversion for 30 min at 4 °C. Insoluble cell debris was subsequently removed by filtration through a 40 μm cell strainer (SPL Life Sciences, 93040). This crude extract was centrifuged at $16,000 \times g$ for 30 minutes to obtain a soluble supernatant fraction for IP. Soluble protein fractions from different strains were diluted to the same concentration prior to IP. DynaBeads were coupled with the anti-HA epitope 3FA antibody (Roche, 11867423001) according to the manufacturer's instructions (Invitrogen,14311D). For each sample, 7.5 mg of the coupled beads were added to 1 ml of the extract and placed on a roller at 4 °C for 1 h. The beads were then washed once with IP buffer containing 0.1% (w/v) Tween-20, twice with IP buffer, once with the manufacturer's LWB buffer supplemented with 0.02% (w/v) Tween-20, and thrice with LWB. Bound proteins were eluted twice according to the manufacturer's instructions, after which, they are combined and dried overnight at room temperature using a vacuum concentrator. Dried proteins were suspended in sodium dodecyl sulfate (SDS) polyacrylamide gel electrophoresis (PAGE) loading buffer. IP with GFP-Trap/RFP-Trap M beads (ChromTech, gtma-10/ rtma-10) was done similarly, except detergents were excluded, and the washed beads and bound proteins were directly suspended in SDS-PAGE loading buffer. Samples were run into the resolving gel by approximately 1 cm after which gels were stained with Coomassie blue, de-stained and extensively washed with de-ionized water. The entire Coomassie staining regions was excised and processed for mass spectrometry. Reduction, alkylation, trypsin digestion and analysis by mass spectroscopy were carried out by the Protein and Proteomics Centre, National University of Singapore. To graphically present the IP results, for each experiment, the Max-Quant signal intensity[76] of each interactant is normalized against the intensity of the IP target to account for differences in IP efficiency. The normalized intensities of interactants in the wild-type IP are then used as reference (i.e, set to 1), to which signal intensities in mutant IPs are compared. These values are represented in a linear grayscale with 0 equal to white and 1 equal to black. Values that exceed 1 are indicated numerically on the figure.

**Native PAGE and Western blotting.** To prepare extracts, frozen N. crassa powder was added to an equal volume of native sample buffer (Invitrogen, BN2008) and thawed on ice for 30 minutes. This mixture was centrifuged at $16,000 \times g$, 4 °C for 30 min and the supernatant was loaded on 3–12% blue native PAGE gels (Invitrogen, BN1001BOX). Electrophoresis was carried out at 4 °C at 100 V for 40 min in dark blue cathode buffer, followed by 150 V for 100 minutes in light blue cathode buffer (Invitrogen, BN2007).

For first dimension blotting, proteins separated by native PAGE were transferred to PVDF membrane with native transfer buffer at 10 V for 10 h at 4 °C. The membrane was de-stained with 100% methanol before it was blocked in tris-buffered saline (10 mM tris pH 7.4, 150 mM NaCl), 0.1% Triton X-100 (TBS-T), 5% nonfat milk (Bio-RAD, #170–6404) for 1 hour at room temperature. The membrane was probed overnight with anti-HA-HRP antibody (#, 1:2500 dilution) in TBS-T, 1% non-fat milk, and imaged with a gel imaging system (ChemiDoc Touch, Bio-RAD). For second dimension blotting, the whole lane of each sample was cut from the BNP gel and soaked in SDS loading buffer for 15 minutes at room temperature. This gel strip is then loaded on the top of an SDS-PAGE gel and sealed with 1% agarose in 125 mM Tris pH 6.8. Standard electrophoresis and blotting procedures were carried out following this step.

**Solution nuclear magnetic resonance (NMR) analysis**. The SHD from *N. crassa* SPA-2 (from glycine-84 to serine-217) was expressed as a HIS-tagged protein in *Escherichia coli* BL21 in M9 medium (12.8 g/L $Na_2HPO_4.7H_2O$, 3 g/L $KH_2PO_4$, 0.5 g/L NaCl, 2 mM $MgSO_4$, 0.1 mM $CaCl_2$, 0.2% $^{13}C$-labeled glucose (Cambridge Isotope Laboratories, Inc, CLM-1396-5) and 0.1% $^{15}N$-labelled $NH_4Cl$ (Sigma-Aldrich, 299251)). The protein was purified under native conditions using Ni-NTA resin (QIAGEN, 30230), followed by gel filtration using Superdex75 column (GE Healthcare). Fractions containing the SHD protein were concentrated to 2 mM in 10 mM phosphate buffer at pH 6.5 with 1 mM EDTA, 1 mM DTT, 0.05% $NaN_3$ and 5% $D_2O$. All NMR experiments were performed on a Bruker Avance 800 spectrometer equipped with a cryo-probe at 25 °C. 2D HSQC, 3D HNCA[77], HNCOCA[78], MQ-(H)CCH-TOCSY[79] and 4D NOESY[80] were recorded using TOPSPIN software (www.bruker.de) without non-uniform sampling scheme. NMR spectra were processed using NMRPipe v10.8[81] and analysed using NMRFAM-Sparky v3.108[82]. Backbone and side-chain resonance assignments were achieved using the 4D NOESY-based strategy[83]. Unambiguous NOEs were obtained from three sub-spectra: $^{13}C,^{15}N$-edited, $^{13}C,^{13}C$-edited, and $^{15}N,^{15}N$-edited 4D NOESY. Distance constraints were obtained from the NOEs assigned, while dihedral angle restraints of φ and ψ were calculated with TALOS+[84] using the assigned chemical shifts of $C_\alpha$, $C_\beta$, N, $H_\alpha$, and HN.

The structure was determined using distance and dihedral angle constraints derived from NOEs and chemical shifts (Supplementary Table 2). Except 15 proline residues, five N–H correlations (M1, R25, N34, K35 and G103) were not observable in the 2D HSQC spectrum and thus could not be assigned. The initial structure calculation was employed with Xplor-NIH[85] using the conventional simulated annealing protocol from an extended conformation of SHD. Then the best folded models with the lowest total energy were selected for EEFX force-field implicit refinement[86] using Xplor-NIH. Both protocols employ the internal variable module and share the same basic scheme: (i) torsion angle dynamics at high-temperature (3,500 K) for 15,000 timesteps; (ii) torsion angle dynamics with simulated annealing, where the temperature is reduced from the initial high temperature value to 25 K in steps of 12.5 K, for a time of 0.4 ps per temperature step (refinement protocol); (iii) 500 steps of Powell torsion angle minimization; and (iv) 500 steps of Powell Cartesian minimization.

In the high temperature stage, experimental dihedral angle restraints and distance restraints were applied with respective force constants of $k_{CDIH} = 10$ kcal $mol^{-1}$ $rad^{-2}$ and $k_{DIST} = 2$ kcal $mol^{-1}$ $rad^{-2}$. In the simulated annealing stage, $k_{CDIH}$ was set to 200 kcal $mol^{-1}$ $rad^{-2}$ and $k_{DIST}$ was increased geometrically from 2 to 30 kcal $mol^{-1}$ $rad^{-2}$. The torsionDB statistical torsion angle potential was included with a force constant set to $k_{tDB} = 0.02$ kcal $mol^{-1}$ $rad^{-2}$ in the high temperature stage and ramped geometrically from 0.02 to 2 kcal $mol^{-1}$ $rad^{-2}$ during simulated annealing.

A total of 100 EEFX force-field refined structures was calculated and 20 conformers with the lowest total energy were deposited with the PBD ID 6LAG and the assigned chemical shifts are deposited with the BMRB ID 36299.

**Bioinformatics**. Sequence conservation shown in Figs. 1b, 4a, 5a,d, and 7c was determined from the multiple sequence alignments which were constructed using MAFFTv6.240[87]. For each position in the alignment, the percentage of amino acids in each of the following groups was calculated: aromatic (phenylalanine, tyrosine, tryptophan), polar (serine, threonine, glutamine, asparagine), negatively charged (aspartic acid, glutamic acid), positively charged (lysine, arginine, histidine), hydrophobic (alanine, valine, leucine, isoleucine), others (glycine, cysteine, methionine, proline). The highest percentage will be used as the conservation score for that particular position. Potential metazoan homologs of fungal SPA-2 were identified by searching the human reference proteome (UP000005640) using an hmm profile constructed from the alignment of fungal SPA-2 sequences (Supplementary Fig. 7). Phylogenetic distribution, and sequence divergence of proteins indicated in Fig. 7b were determined from branch lengths of maximum likelihood trees inferred with RAxML v8.1.15[88] using input alignment constructed with MAFFT v6.240 and trimmed with TrimAl v1.1[89] as previously described[35]. Individual protein domains shown in Fig. 7b correspond to the following regions of the *N. crassa* sequences, SPA-2 SHD: amino acids 118–180, SPA-2 FAT: amino acids 753–873 (Unitprot ID: V5IQM7). CCP-1 Pezizomycotina-specific coiled-coil containing region: amino acids 274-709. All proteomes used in these analyses are listed in Supplementary Table 3.

**Statistics and Reproducibility**. When representative images are shown (Fig. 1c, e, f; 2a; 3b, e, i; 5c, d, f; 6a–c and supplementary fig. 4b–e), five independent hyphae were imaged and one representative image is shown. The full dataset is available in the Source data file. Error bars in Figs. 1d, 5e and Supplementary Fig. 4a represent mean ± SD. Each measurement was made independently at least three times.

**Reporting summary**. Further information on research design is available in the Nature Research Reporting Summary linked to this article.

## Data availability

A reporting summary for this Article is available as a Supplementary Information file. The source data underlying Figs. 1c–f; 2a–c; 3b, e, f, h, i; 5b–d, f; 6a–c and Supplementary Figs. 1a, b; 2a–c and 4a–e are provided in the Source data file. The *N. crassa* SPA-2 Spa Homology Domain (SHD) NMR structure is deposited with the Protein Data Bank under accession number PBD 6LAG [https://doi.org/10.2210/pdb6LAG/pdb] and the assigned chemical shifts are deposited with the Biological Magnetic Resonance Bank under accession number BMRB ID 36299 [https://doi.org/10.13018/BMR36299]. Other data supporting the findings of this manuscript are available from the corresponding author upon request. Source data are provided with this paper.

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

## Acknowledgements

P.Z., T.A.N., J.Y.W., M.L. and G.J. are supported by the Temasek Life Sciences Laboratory and Singapore Millennium Foundation. D.W.Y. and J.S.F. are supported by the Singapore Ministry of Education Tier 1 grant #R-154-000-A50-114. We thank Kristina Smith for critical reading of the manuscript.

## Author contributions

G.J. conceived the project. P.Z., J.Y.W., M.L., and T.-A.N. performed experiments. T.A.N. performed bioinformatics analysis. D.W.Y. and J.S.F. performed the NMR structure analysis. G.J. wrote the manuscript with input from all authors.

## Competing interests

The authors declare no competing interests.
