## [Peer Review File · Nature Communications]

Reviewers' comments:

Reviewer #1 (Remarks to the Author):

A manuscript submitted by Zheng et al. describes identified interaction cascades by two SPK scaffolds which lead to effector recruitment and SPK-accumulation of secretory vesicles does not seem to be necessitated. Among them, the conserved SPA-2 Spa homology domain (SHD) recruits an effector accelerating SPK F-actin stabilization. The author determined the SHD structure and the amphipathic groove looks to be important for binding the effector. This reviewer is asked to provide comments for the structural quality particularly.

Page 16 Line 31 - Page 17 Line 3.

Most NMR experiments and analyses are conducted by computer software packages and pulse programs. Providing the information about what kind of software packages to acquire and analyze the data is crucial because the results will vary significantly. The author only references MQ-(H)CCH-TOCSY and 4D NOESY and it lacks pulse schemes and whether non-uniform sampling applied or not. NMRPipe and Sparky are not referenced while 4D NOESY-based strategy is referenced which is not recommended because it does not credit software packages properly while the author only promotes their strategies.

Page 17 Line 3-8

There is no such a standard simulated annealing method. Xplor-NIH is highly sophisticated versatile molecular dynamics program and the author should detail structure calculation steps applied. Also, the protein is all alpha helical structure which inherently exhibits very small number of long-range restraints. The author should provide full information about assignments and restraints used. To maximize the structural quality, RDC should be experimented and used to cross-validate the structure because this is dynamic helical structure and the author uses the structure to support their hypothesis on effector recruitment. Also, BMRB entry number has not been provided.

Reviewer #2 (Remarks to the Author):

Fungi are a critically important kingdom of organisms from agents of disease to agricultural pests, to uses for industrial production of enzymes. All of these phenomenon require the production of a specialized cell type, the hyphae, a polarized cell unique to fungi. Yet, how hyphal growth is achieved remains largely unsolved. Here the authors provide a remarkable set of experiments that establish not just one, but two unique scaffolding systems associated with the assembly of the Spitzenkorper, a tip associated 'organelle' intimately associated with hyphal growth. Their experiments are carefully designed and the interpretations of their results are well founded. This study represents a major step forward in our understanding of hyphal growth and will be of broad interest to fungal biologists, developmental and cell biologists alike. I have no major criticisms of the work. A few minor considerations are listed and none of them should be a barrier to the acceptance of the manuscript.

General Comments:

- It is unclear which organism is used in this study until the results section. It would be helpful if the authors could either mention at some point that this study is performed using *N. crassa*, or add the initials of each organism to the gene/protein names discussed in the introduction (this would be particularly helpful when the text switches back and forth between yeast and filamentous

fungi).

- There are a few acronyms in the introduction that could be defined the first time they are mentioned (for example: NDR, GIT/PIX).
- Throughout the manuscript, *Neurospora* is consistently not italicized. Also, if the species used was *N. crassa*, I think it would be best that the authors specify (for example, p.7, line 21 could say "...mutation in *N. crassa* SPA-2...").

Minor Revisions:

- The sentence ending on line 17 needs a period after the references.
- It is unclear what the first sentence on page 4 referring to. As it is currently worded, it seems like it is in reference to the sentence mentioning the requirement of Pea2 for SPA-2 tip-localization (however, it would make more sense if it is actually referring to the polarisome). The authors could consider simply rewording this to clarify.
- P. 4, Line 3 – both *Aspergillus* and *Neurospora* should be italicized. It would also be helpful to specify which species is being discussed, and to include the protein name(s) for each particular organism. For example, the text could read "BudA/BUD-6" and "SpaA/SPA-2" (assuming "Aspergillus" is in reference to *A. nidulans*).
- There is quite a bit of information in the results that could potentially be moved to the discussion (for example, the paragraph at the top of page 6 discusses possible reasonings for the weak localization results provided).
- P. 9, line 5 – the abbreviation SPK could be used here.
- P. 12, line 26-30 – a reference could be provided here.
- The experimental comparison to the coronin mutant was not mentioned at all in the results. Instead, it is lightly touched on in the discussion (p.12, line 25; Supplementary Fig. 7).
- P. 12, line 31 – again, both genus names should be italicized and the species name should be provided.
- P. 15, line 1 – the parenthetical statement beginning here should be closed at some point.
- Journal names and abbreviations are not consistent in the references.

Minor Corrections in Figures & Supplementary Information:

- Fig. 1d – does "three independent measurements" mean that three hyphae were measured for each strain? Also, it is difficult to see the standard deviation markers because the bars are also solid black.
- Fig. 3e & 3i – It would be helpful if the authors listed the protein name with these figures, as they did in Fig. 3b for SPZ-1.
- *Neurospora* should be italicized (and the species used should be listed as well) in all of the figure names and legends.
- It would be helpful to again define the SHD acronym in Supplementary Fig. 3 once so that it can stand alone.
- Supplementary Fig. 4a – it is difficult to see the standard deviation markers here as well.
- References for Supplementary Figures are inconsistent (for example, reference 2 does not have a journal listed, page numbers are sometimes abbreviated and sometimes fully written out, websites are in different formats, etc.).

Reviewer #3 (Remarks to the Author):

In this original research article, the authors demonstrate that SPZ-1, a coiled-coil Spitzenkörper (SPK) protein that was previously identified in multicellular Ascomycota, acts as a cargo adaptor of MYO-5 to transport two different scaffold complexes to the SPK.

The article is carefully and very well written, the experiments elegantly designed and performed and the figures of excellent quality.

I include below a few minor questions for the authors to address and some suggestions that could contribute to the final version of this article.

Title: taking into account that the secretory vesicles that accumulate at the SPK do not depend on either of the proteins of the two scaffold complexes identified in this study, and considering previous studies by other authors, I would ask the authors to consider rephrasing the title and consider the SPK and the tip polarity apparatus as different entities. This and other studies show that although several polarity related proteins localize at the SPK, the SPK is not just a polarity apparatus. What about?:

1. Modular assembly and ancient evolutionary origin of the hyphal Spitzenkörper and the tip polarity apparatus.

or

2. Modular assembly and ancient evolutionary origin of the hyphal tip growth apparatus. This second option would account for the SPK, the polarisome and related polarity proteins, scaffold proteins, etc.

Abstract: see comment above. Delete "polarity apparatus"

Results:

P6. L21-23. The SPZ-1 deletion variant lacking region 3 retains only weakly its interaction with MYO-5 and while it localizes at the tip, it seems to do so in a reduced area compared to the WT variant. Please adjust text to describe in more detail.

P7.

L9. Spa-2 should be Spa2.

L21. How do the authors establish that L132A in Neurospora SPA-2 leads to a full loss-of-function? Figure shows a growth rate reduction of 30%.

L30. Authors describe that L132A mutation abolishes the ability of SPA-2 to bind CPP-1. What about the weak band in Fig. 5b?

P8.

L1. "Therefore, we next examined the impact of ccp-1 deletion on actin filaments". Please delete filaments. It is not really possible to observe the actin filaments in Fig. 5d. One can see the Spk actin and in some panels the actin patches in the subapical region of the cell surface.

L5. "Interestingly, CCP-1 loss-of-function also impairs SPK incorporation of LAH-2, but does not affect tip-localization of SPZ-1, SPA-1 or JNS-1" add (Fig. 5f) at the end of this result, instead of at the end of the paragraph.

Discussion:

P9. Change Spitzenkörper polarity apparatus to tip polarity apparatus.

L5. C

L11. Binds should be bind.

References:

Please cite in text where appropriate and add to reference list:

PMID: 18216285

Mol Biol Cell. 2008 Apr;19(4):1439-49. doi: 10.1091/mbc.E07-05-0464.

The tip growth apparatus of *Aspergillus nidulans*.

Taheri-Talesh N, Horio T, Araujo-Bazán L, Dou X, Espeso EA, Peñalva MA, Osmani SA, Oakley BR.

PMID: 15701784

Eukaryot Cell. 2005 Feb;4(2):225-9.

Polarisome meets Spitzenkörper: microscopy, genetics, and genomics converge.

Harris SD, Read ND, Roberson RW, Shaw B, Seiler S, Plamann M, Momany M.

Author information

Suppl. Fig. 4 b and c are not cited nor described in main text of article.

We thank the reviewers for helping us improve our manuscript. Our responses are found below highlighted in grey.

Reviewers' comments:

Reviewer #1 (Remarks to the Author):

A manuscript submitted by Zheng et al. describes identified interaction cascades by two SPK scaffolds which lead to effector recruitment and SPK-accumulation of secretory vesicles does not seem to be necessitated. Among them, the conserved SPA-2 Spa homology domain (SHD) recruits an effector accelerating SPK F-actin stabilization. The author determined the SHD structure and the amphipathic groove looks to be important for binding the effector. This reviewer is asked to provide comments for the structural quality particularly.

Page 16 Line 31 - Page 17 Line 3.

Most NMR experiments and analyses are conducted by computer software packages and pulse programs. Providing the information about what kind of software packages to acquire and analyze the data is crucial because the results will vary significantly. The author only references MQ-(H)CCH-TOCSY and 4D NOESY and it lacks pulse schemes and whether non-uniform sampling applied or not. NMRPipe and Sparky are not referenced while 4D NOESY-based strategy is referenced which is not recommended because it does not credit software packages properly while the author only promotes their strategies.

All the experimental methods are now detailed and referenced.

Page 17 Line 3-8

There is no such a standard simulated annealing method. Xplor-NIH is highly sophisticated versatile molecular dynamics program and the author should detail structure calculation steps applied. Also, the protein is all alpha helical structure which inherently exhibits very small number of long-range restraints. The author should provide full information about assignments and restraints used. To maximize the structural quality, RDC should be experimented and used to cross-validate the structure because this is dynamic helical structure and the author uses the structure to support their hypothesis on effector recruitment. Also, BMRB entry number has not been provided.

1. All the steps used for structure calculation are now included in the Materials and Methods. The restraints used and structural statistics are listed in the new Supplementary Table 3.

2. A large number of long-range NOEs (178) defined the relative orientation of helices. Thus, additional RDC data were not necessary for cross-validating the structure.

3. The BMRB ID is now provided along with the PDB ID.

Reviewer #2 (Remarks to the Author):

Fungi are a critically important kingdom of organisms from agents of disease to agricultural pests, to uses for industrial production of enzymes. All of these phenomenon require the

production of a specialized cell type, the hyphae, a polarized cell unique to fungi. Yet, how hyphal growth is achieved remains largely unsolved. Here the authors provide a remarkable set of experiments that establish not just one, but two unique scaffolding systems associated with the assembly of the Spitzenkorper, a tip associated 'organelle' intimately associated with hyphal growth. Their experiments are carefully designed and the interpretations of their results are well founded. This study represents a major step forward in our understanding of hyphal growth and will be of broad interest to fungal biologists, developmental and cell biologists alike. I have no major criticisms of the work. A few minor considerations are listed and none of them should be a barrier to the acceptance of the manuscript.

General Comments:

- It is unclear which organism is used in this study until the results section. It would be helpful if the authors could either mention at some point that this study is performed using *N. crassa*, or add the initials of each organism to the gene/protein names discussed in the introduction (this would be particularly helpful when the text switches back and forth between yeast and filamentous fungi).

We refer to the model system at the beginning of the last paragraph of the introduction and again in the first sentence of the results section. We have modified the introduction to clarify which species are being referred to as we introduce the various proteins.

- There are a few acronyms in the introduction that could be defined the first time they are mentioned (for example: NDR, GIT/PIX).

The full-length names for acronyms are now given at their first occurrence.

- Throughout the manuscript, *Neurospora* is consistently not italicized. Also, if the species used was *N. crassa*, I think it would be best that the authors specify (for example, p.7, line 21 could say "...mutation in *N. crassa* SPA-2...").

We have used the full species name, *Neurospora crassa* at first usage and *N. crassa* thereafter.

Minor Revisions:

- The sentence ending on line 17 needs a period after the references.

The sentence has been corrected.

- It is unclear what the first sentence on page 4 referring to. As it is currently worded, it seems like it is in reference to the sentence mentioning the requirement of Pea2 for SPA-2 tip-localization (however, it would make more sense if it is actually referring to the polarisome). The authors could consider simply rewording this to clarify.

The sentence has been revised for greater clarity.

- P. 4, Line 3 – both *Aspergillus* and *Neurospora* should be italicized. It would also be helpful to specify which species is being discussed, and to include the protein name(s) for each particular organism. For example, the text could read "BudA/BUD-6" and "SpaA/SPA-2" (assuming "Aspergillus" is in reference to *A. nidulans*).

The suggested edits have been made.

- There is quite a bit of information in the results that could potentially be moved to the discussion (for example, the paragraph at the top of page 6 discusses possible reasonings for the weak localization results provided).

In the passage cited we are discussing an interpretive caveat, which we would like to address when it first arises. We feel that this is more effective in this case as opposed to bringing it up again in the discussion, where it would disrupt the narrative flow.

- P. 9, line 5 – the abbreviation SPK could be used here.

This is the first major statement of the discussion, so we prefer to revert here to the full name.

- P. 12, line 26-30 – a reference could be provided here.

A reference to the initial growth rate measurements has been added.

Francis J. Ryan, G. W. Beadle and E. L. Tatum *American Journal of Botany* Vol. 30, No. 10 (Dec., 1943), pp. 784-799

- The experimental comparison to the coronin mutant was not mentioned at all in the results. Instead, it is lightly touched on in the discussion (p.12, line 25; Supplementary Fig. 7).

The paper is primarily driven by the discoveries relating to protein complexes, which appear to be primarily related to attaining maximal growth rates. However, the question of a role in morphogenesis arises based on the function of Polarisome components in budding yeast. We feel that bringing these results up in the discussion is effective as it lets us present the main results in an uninterrupted manner.

- P. 12, line 31 – again, both genus names should be italicized and the species name should be provided.

The corrections have been made.

- P. 15, line 1 – the parenthetical statement beginning here should be closed at some point.

The correction has been made.

- Journal names and abbreviations are not consistent in the references.

These have been corrected.

Minor Corrections in Figures & Supplementary Information:

- Fig. 1d – does “three independent measurements” mean that three hyphae were measured for each strain? Also, it is difficult to see the standard deviation markers because the bars are also solid black.

The growth rate is made using the race tube method (3 replicates per strain), so we are measuring the advancement of the growth front, not the growth of individual hyphae. The original reference to the method of growth rate measurement has been added.

The column and standard deviation are shown in grey and black, respectively for more visibility.

- Fig. 3e & 3i – It would be helpful if the authors listed the protein name with these figures, as they did in Fig. 3b for SPZ-1.

The figure has been amended accordingly.

- *Neurospora* should be italicized (and the species used should be listed as well) in all of the figure names and legends.

The corrections have been made.

- It would be helpful to again define the SHD acronym in Supplementary Fig. 3 once so that it can stand alone.

The acronym is now defined in the legend.

- Supplementary Fig. 4a – it is difficult to see the standard deviation markers here as well.

The columns and standard deviation markers are shown in grey and black, respectively for more visibility.

- References for Supplementary Figures are inconsistent (for example, reference 2 does not have a journal listed, page numbers are sometimes abbreviated and sometimes fully written out, websites are in different formats, etc.).

The references have been edited to provide the missing information.

Reviewer #3 (Remarks to the Author):

In this original research article, the authors demonstrate that SPZ-1, a coiled-coil Spitzenkörper (SPK) protein that was previously identified in multicellular Ascomycota, acts as a cargo adaptor of MYO-5 to transport two different scaffold complexes to the SPK.

The article is carefully and very well written, the experiments elegantly designed and performed and the figures of excellent quality.

I include below a few minor questions for the authors to address and some suggestions that could contribute to the final version of this article.

Title: taking into account that the secretory vesicles that accumulate at the SPK do not depend on either of the proteins of the two scaffold complexes identified in this study, and considering previous studies by other authors, I would ask the authors to consider rephrasing the title and consider the SPK and the tip polarity apparatus as different entities. This and other studies show that although several polarity related proteins localize at the SPK, the SPK is not just a polarity apparatus. What about?:

1. Modular assembly and ancient evolutionary origin of the hyphal Spitzenkörper and the tip polarity apparatus.

or

2. Modular assembly and ancient evolutionary origin of the hyphal tip growth apparatus. This second option would account for the SPK, the polarisome and related polarity proteins, scaffold proteins, etc.

Abstract: see comment above. Delete “polarity apparatus”

We agree that the combined use of the term polarity apparatus and Spitzenkörper is likely to be confusing to readers. The revised manuscript no longer combines these terms. Our view is that the SPK is most accurately defined as being comprised of both vesicular and proteinaceous constituents. We make this definition in the first sentence of the abstract “The Spitzenkörper (SPK) constitutes a collection of secretory vesicles and polarity-related proteins intimately associated with the tip growth of fungal hyphae.”

We have revised the title to: **“Spitzenkörper assembly mechanisms reveal conserved features of fungal and metazoan cell polarity scaffolds”**.

Results:

P6. L21-23. The SPZ-1 deletion variant lacking region 3 retains only weakly its interaction with MYO-5 and while it localizes at the tip, it seems to do so in a reduced area compared to the WT variant. Please adjust text to describe in more detail.

We have revised the presentation to mention that this variant has lower steady-state protein levels compared with the other tagged strains. This is likely to account for the diminished pull-down and weaker signal at the hyphal tip.

P7.

L9. Spa-2 should be Spa2.

The correction has been made.

L21. How do the authors establish that L132A in Neurospora SPA-2 leads to a full loss-of-function? Figure shows a growth rate reduction of 30%.

The SPA-2 deletion growth rate is definitive of a full loss-of-function. The L132A mutant has a growth rate slightly slower than the deletion mutant. This could reflect a weak dominant negative effect, but could also be due to background genetic variation. At this junction we cannot distinguish between these possibilities.

L30. Authors describe that L132A mutation abolishes the ability of SPA-2 to bind CPP-1. What about the weak band in Fig. 5b?

The observation is correct. We have amended the text to replace the term “abolishes” with “significantly diminishes”.

P8.

L1. “Therefore, we next examined the impact of ccp-1 deletion on actin filaments”. Please delete filaments. It is not really possible to observe the actin filaments in Fig. 5d. One can see the Spk actin and in some panels the actin patches in the subapical region of the cell surface.

The suggested edit has been made.

L5. “Interestingly, CCP-1 loss-of-function also impairs SPK incorporation of LAH-2, but does not affect tip-localization of SPZ-1, SPA-1 or JNS-1” add (Fig. 5f) at the end of this result, instead of at the end of the paragraph.

The suggested edit has been made.

Discussion:

P9. Change Spitzenkörper polarity apparatus to tip polarity apparatus.

L5.

We have changed the wording to “proteinaceous Spitzenkörper scaffolds”.

L11. Binds should be bind.

The sentence refers to the paper’s results and not SHDs in general, so it has been revised to “The SPA-2 SHD binds the F-actin effector CCP-1”. Also, we realize that because SHD is the acronym for “Spa homology domain”, that we need to delete domain from the passage.

References:

Please cite in text where appropriate and add to reference list:

PMID: 18216285

Mol Biol Cell. 2008 Apr;19(4):1439-49. doi: 10.1091/mbc.E07-05-0464.

The tip growth apparatus of *Aspergillus nidulans*.

Taheri-Talesh N, Horio T, Araujo-Bazán L, Dou X, Espeso EA, Peñalva MA, Osmani SA, Oakley BR.

PMID: 15701784

Eukaryot Cell. 2005 Feb;4(2):225-9.

Polarisome meets Spitzenkörper: microscopy, genetics, and genomics converge.

Harris SD, Read ND, Roberson RW, Shaw B, Seiler S, Plamann M, Momany M.

Author information

The suggested citations have been added.

Other changes to the manuscript.

1. In the initial submission, the dependency of GYP-3 on SPA-2 for tip-localization was based on a recent citation. During the review period, we obtained evidence showing GYP-3 tip-localization is abolished in the SHD point mutant. The new information is mentioned in the results and shown in Supplementary figure 4e.

2. Data availability is added to the end of the paper.

3. Reference numbers are edited accordingly and coloured in blue.

4. Other minor edits and corrections have been made and highlighted in grey.

Reviewers' comments:

Reviewer #1 (Remarks to the Author):

I appreciate the fact that the authors put efforts on revision. My remaining concern is that structure quality from the wwPDB validation report is not enough.

According to the revised methods, it does not implement refinement step after gradient minimization steps. To improve, I recommend using the water refine tool for explicit refinement or using EEFX force-field for implicit refinement which looks mandatory now. I doubt why detailed calculation protocol is available ONLY upon request which it is not really complicated to detail and also can be in supplementary information.

Backbone r.m.s.d. for detected ordered regions is above 1A from pg 3 in wwPDB validation report and it is high and there are 5 clusters detected which means a large number of long-range NOEs (178) hardly define the relative orientation in high fidelity manner not to mention clash score is above 30 and many clashes still exist.

Torsion angles, there are 5% outliers and percentile is not great either.

I hope the authors conduct more refinement on structure and use a larger number sampling to accommodate these issues.

RE: Zheng et al. Spitzenkörper assembly mechanisms reveal conserved features of fungal and metazoan polarity scaffolds

We thank the reviewer for taking the time to help us improve the manuscript. Please see our responses below, which are highlighted in grey. We believe that the refined NMR structure fully supports the paper's conclusions.

Reviewer #1 (Remarks to the Author):

1. I appreciate the fact that the authors put efforts on revision. My remaining concern is that structure quality from the wwPDB validation report is not enough.

We checked our NOESY spectra again, and assigned additional 34 long-range NOEs. With these NOEs and further refinement by using the EEFX force field, we have improved the structure quality significantly. Please see the new PDB validation report for documentation of the structure's improvement.

2. According to the revised methods, it does not implement refinement step after gradient minimization steps. To improve, I recommend using the water refine tool for explicit refinement or using EEFX force-field for implicit refinement which looks mandatory now.

We have implemented the EEFX force field method with good results.

3. I doubt why detailed calculation protocol is available ONLY upon request which it is not really complicated to detail and also can be in supplementary information.

All procedures have been documented and are supplied with the manuscript (please refer to page 17 in the revised version). Thus, no request for additional information is necessary.

4. Backbone r.m.s.d. for detected ordered regions is above 1Å from pg 3 in wwPDB validation report and it is high and there are 5 clusters detected which means a large number of long-range NOEs (178) hardly define the relative orientation in high fidelity manner not to mention clash score is above 30 and many clashes still exist.

A number of improvements are documented in the PDB report.

1. The RMSD is now 0.89Å.
2. There are now 4 clusters instead of 5.
3. The clashscore is significantly improved from 17 to 1.

5. Torsion angles, there are 5% outliers and percentile is not great either.

1. The percentiles score has gone from 4 (all PDB) and 24 (all NMR), to 35 and 77, respectively. The initial structure had 27 unique outliers (5%) compared to 4 (0.7%) in the revised structure.
2. We further note that the protein sidechains percentiles scores have improved from 21 (all PDB) and 68 (all NMR), to 73 and 96, in the revised structure.

Notes and other changes

1. The N-terminal His-tag and spacing sequences are included in the updated PDB validation report. Thus, the sequence length is increased from 135 to 155.
2. There was an error with numbering of the *N. crassa* SPA-2 SHD domain. The L to A mutation in SPA-2 SHD region is corrected from L132A to L133A.
3. We have added a recent reference to the introduction which documents an additional component of the budding yeast polarisome (Reference 29, last 3 lines of page 3)
4. The highlighting in the main manuscript is restricted to changes associated with the NMR structure.
5. The abstract and discussion has been edited for better narrative flow. Conclusions are unchanged.
6. When referring to our results, we have replaced the term “tip-localization” with “SPK-localization” or “SPK-residency” which we believe are more specific and accurate terms.

REVIEWERS' COMMENTS:

Reviewer #1 (Remarks to the Author):

All my concerns have been considered and authors have made significant improvements on the structure and manuscript. I am fully satisfied.